# Effect of Mobile Phone Addiction on Physical Exercise in University Students: Moderating Effect of Peer Relationships

**DOI:** 10.3390/ijerph20032685

**Published:** 2023-02-02

**Authors:** Yahui Han, Guoyou Qin, Shanshan Han, Youzhi Ke, Shuqiao Meng, Wenxia Tong, Qiang Guo, Yaxing Li, Yupeng Ye, Wenya Shi

**Affiliations:** 1Institute of Sports Science, Kyunggi University, Suwon 449701, Republic of Korea; 2Physical Education Institute, Hanjiang Normal University, Shiyan 442000, China; 3Institute of Sports Science, Nantong University, Nantong 226019, China; 4School of Physical Education, Shanghai University of Sport, Shanghai 200438, China; 5Physical Education College, Yangzhou University, Yangzhou 225127, China; 6Physical Education College, Shangqiu University, Shangqiu 476000, China; 7School of Physical Education, Jing-Gang-Shan University, Ji’an 343009, China; 8Physical Education College, Guangxi Minzu Normal University, Chongzuo 532200, China

**Keywords:** addictive behavior, physical exercise, peer support, university student, sports involvement, impact mechanism

## Abstract

Objective: The influence of mobile phone addiction (MPA) on physical exercise in university students was explored, and peer relationships were introduced as a moderating variable. Methods: A cross-sectional study design was adopted, and an online survey questionnaire was conducted to investigate two universities in Nantong City, Jiangsu Province, and Chongzuo City, Guangxi Zhuang Autonomous Region. A total of 4959 university students completed the questionnaire. Measurement tools included the Mobile Phone Addiction Tendency Scale, the Physical Activity Rating Scale, and the Peer Rating Scale of university students. Results: University students scored 39.322 ± 15.139 for MPA and 44.022 ± 7.735 for peer relationships, with 87.8% of their physical exercise, in terms of exercise grade, being classified as medium or low intensity. The MPA of the university students was negatively correlated with peer relationships (r = −0.377, *p* < 0.001) and physical exercise behavior (r = −0.279, *p* < 0.001). The moderating effect of peer relationships on the MPA-physical exercise behavior relationship was significant (ΔR^2^ = 0.03, *p* < 0.001). Conclusions: The physical exercise of university students was at a medium or low intensity. The more serious the university students’ addiction to mobile phones was, the lower the amount of physical exercise. The physical activity of males was higher than that of females. MPA and peer relationships were the limiting factors of the physical exercise behavior of university students. Under the lower effect of peer relationship regulation, MPA had a greater negative impact on physical exercise behavior. The data from this research can provide theoretical support to improve the participation of university students in physical activities.

## 1. Introduction

Mobile phone addiction (MPA) not only reduces sleep quality and produces negative emotions, such as burnout and procrastination [1,2,3], but also increases screen time and sedentary behavior, directly and indirectly interfering with the various forms of physical activities of university students [1,4,5,6]. Obesity, cardiovascular disease, and decreased immunity are some of the issues caused by low physical activity engagement [7,8]. Smartphone use has shown strong penetration and impact among university students [9]. According to the 47th Statistical Report on the Development of Chinese Public Internet Networks [10], mobile Internet users in China totaled 986 million as of December 2020, among whom 21.0% were students. Although MPA indeed triggers a series of negative living habits in university students, the effect of MPA on physical activity among those with behavioral self-management abilities needs to be investigated. On the one hand, physical and mental health problems caused by a decline in physical activity are more prevalent among university students [11,12]. On the other hand, university students, as individuals with self-management abilities, have just reached adulthood, so addictive behaviors, such as mobile phone addiction, may be more harmful [13,14]. Exploring the intrinsic relationship between MPA and physical activity is of great significance for developing healthy lifestyles in university students.

MPA is a negative psychological and behavioral state in which individuals use mobile phones excessively and frequently in non-study and non-work conditions [15,16]. The characteristics of MPA include excessive/uncontrolled use of mobile phones regardless of the external environment, neglect of real life and mental overdependence on mobile phones, and experiencing withdrawal from using mobile phones accompanied by anxiety and a sense of loss, among others [16,17]. MPA reduces the self-identity and self-harmony of university students and affects their cognition, attitude, decision-making, expression, and experience of social behavior [18]. University students with MPA tend to be addicted to activities, such as acquiring Internet information and tinkering with technical operations in which mobile phones are used as a medium. Correspondingly, their physical activity time, opportunities, and resources are taken up and replaced by excessive screen time or sedentary behavior [19,20]. Therefore, MPA is often accompanied by weak motivation and interest in exercise, described in past studies as “low frequency, short duration, and low intensity,” with the physical activity of university students hardly meeting the ideal recommended amount [7].

Peer relationships refer to the psychological relationships between peers or among peers, and they are spontaneously formed because of common interests, needs, and attitudes. Peer relationships also play a unique and irreplaceable role in the academic development and social adaptation of university students. Good peer relationships are important for the development of social competence, which is a relevant source of social needs, social support, and security. Furthermore, peer experience is conducive to the development of the self-concept and personality of university students. The social environment affects an individual’s ability to self-adjust, and it exhibits behavioral paradigms and characteristics [21]. The psychology community has attempted to explore the influential mechanisms of the physical activities of university students and has found that negative interpersonal environmental perception is a limiting factor of physical activity [22]. University students who perceive interpersonal issues experience psychological reactions, such as anxiety, loneliness, and shyness, among others, thus affecting their social adaptation and physical exercise practice. Moreover, interpersonal distress causes people to suffer from self-worth crises, reject positive social behavior, and poorly engage in physical activity.

The influence path of peer relationships on physical exercise can basically be summarized into two modes: positive and negative. When friends have an impact on physical exercise, these two different nature effects may occur together. A positive peer relationship mainly refers to peers supporting participation in physical exercise, including integration and companionship (such as participating in sports activities together), emotional support (such as encouraging behavior), information guidance, device support, etc. [23,24]. A negative peer relationship mainly refers to the adverse effects of peer injury behavior on physical exercise. The direct manifestation of harmful behavior is physical or psychological harm. The occurrence of physical violence during physical exercise, or being ridiculed, satirized, criticized, etc., will not only reduce the fun of participation but also reduce the degree of participation in exercise [25,26].

Peer relationships are a trigger for MPA. Social adjustment theory holds that the interpersonal atmosphere has important radiating effects on human behavior [27]. When people perceive interpersonal trouble, they will likely negatively evaluate the environment and people surrounding them, which not only reduces their social cognitive ability and social adaptability but also increases their dependence on and addiction to mobile phone use. Compared with real social networking—as mobile online social networking has the advantages of “reducing more social clues”, “avoiding being directly evaluated by others”, and “reducing the level of real social anxiety”—individuals who often feel troubled by interpersonal relationships in real life habitually “empathize” with the virtual world and acquire self-esteem maintenance and emotional catharsis through mobile online social networking [17]. In reality, a lack of interpersonal security or interpersonal distress leads to MPA, which can tempt people to shift their attention to virtual networks and seek psychological comfort, emotional support, and care [28].

On the basis of the research background presented above, this study explores the following questions through empirical investigation: (1) What are the current characteristics of MPA, physical exercise behavior, and peer relationships of university students? (2) Do MPA and peer relationships affect university students’ physical exercise? (3) Do peer relationships have a moderating effect when MPA affects the physical exercise of university students? (Figure 1).

## 2. Materials and Methods

This research adopted a cross-sectional research design, the key element of which was the participation of physical education teachers in the research implementation. We asked university students to participate in the survey in the form of an online questionnaire.

### 2.1. Participants

Cluster random sampling was used as the basis of a survey questionnaire involving middle school students from Nantong University (Nantong City, Jiangsu Province) and Guangxi Normal University for Nationalities (Chongzuo City, Guangxi Zhuang Autonomous Region); the survey was conducted in September 2022. The unified use mode of the Questionnaire Star software was used to send the questionnaires, and a total of 5980 questionnaires were distributed. “Filling time not between 250–800 s”, “lack of frequency or time data of any intensity physical activity”, and “reverse question test”, among others, were used as the judging criteria for questionnaire invalidation. Questionnaires returned with key information (e.g., age) missing were eliminated. Finally, 4959 valid questionnaires were obtained, with an effective rate of 82.9%. The distribution of survey respondents is shown in Table 1.

This study sets out criteria for the inclusion and exclusion of university student survey participants. The teacher presented the specifics of the questionnaire before students filled it out and explained everything at the beginning of the questionnaire. The inclusion criterion was university students with student status from the Ministry of Education of China. The exclusion criteria were as follows: (1) university students who were not in school during the implementation of the questionnaire completion stage; (2) university students who were unable to participate in physical exercise due to heart disease, organic disease, or mental illness; (3) university students who did not have student status at the Ministry of Education of China.

The calculation of the minimum sample size was completed using Formula (1) [29], where type I error *α* was set to 0.05, the allowable error, *δ*, was set to 0.01, and the sample rate, *ρ*, was set to 0.05. Given a total of 58,962 people (data updated in 2022), the minimum sample size required for this study was calculated to be 1771. The sampling in this study met the minimum sample size requirement.
(1)$$n=\frac{{\left(\frac{Z\alpha }{\delta }\right)}_{}^{2}*p*(1-p)}{1+[{\left(\frac{Z\alpha }{\delta }\right)}_{}^{2}*p*(1-p)]/N}$$

### 2.2. Measurements

#### 2.2.1. Mobile Phone Addiction Tendency Scale (MPATS)

The MPATS was compiled by the Chinese scholar, Xiong [30]. The MPATS includes four dimensions, namely, withdrawal symptoms (experiencing negative physical or psychological reactions when not participating in mobile phone activities), highlighting behavior (the mobile phone as the center of thinking and behavioral activities), social comfort (the role of mobile phone use in interpersonal communication), and mood changes (emotional changes caused by mobile phones), totaling 16 questions (refer to Appendix A). Each question adopts the five-point Likert scoring method, from “completely non-conforming”, “not very conforming”, “general”, “somewhat conforming”, and “completely conforming”, in order from 1 to 5 points. The highest attainable score is 80 points, and the lowest possible score is 16 points. The higher the score is, the greater the MPA tendency, and vice versa (i.e., the lower the addictive tendency). The load of the four scale factors is between 0.51 and 0.79, and the cumulative variance contribution rate is 54.3%. The results of the confirmatory factor analysis indicate the suitability of using the four-factor scale model. The Cronbach’s α coefficient of the aggregate table is 0.83, and the α coefficients of the four factors are 0.55–0.80; furthermore, the retest reliability of the aggregate table is 0.91, and the retest reliability values for the four factors are 0.75–0.85 [30]. The MPATS is suitable for measuring MPA in Chinese university students.

#### 2.2.2. Physical Activity Rating Scale (PARS-3)

The PARS-3 was compiled by the Japanese scholar, Kōo Hashimoto, and revised by the Chinese scholar, Liang [31]. The PARS-3 examines the amount of physical activity from three aspects, namely, the intensity of physical exercise, the frequency of physical exercise, and the amount of workout time. It also measures the level of physical activity participation (refer to Appendix A). Each problem statement in the PARS-3 is divided into five grades, scored from 1 to 5, and calculated using Equation (2). The obtained scores can be standardized as follows: low exercise (≤19 points), medium exercise (20–42 points), and high exercise (≥43 points). The PARS-3 results are a measure of physical activity, which to a certain extent, can reflect the behavioral status of university students toward sports participation at a specific time. The retest reliability of this scale is 0.820 [31].
Physical exercise volume score = intensity × (time − 1) × frequency(2)

#### 2.2.3. Peer Rating Scale (PRS)

Asher’s PRS [32] was used in this research. The PRS consists of 16 questions on a four-level rating scale. The options on the scale range from “complete conformity” to “completely nonconformant”, assigning values of 1–4 points. For the reverse question conversion, the total score of the 16 questions is subsequently calculated for each student. The higher the score, the better the peer relationship. Previously, Zhang [33] divided the original scale into three factors based on the results of factor analysis: “welcome” (Cronbach’s α = 0.722), “autism” (Cronbach’s α = 0.749), and “rejection” (Cronbach’s α = 0.743). The overall internal consistency of the scale at Cronbach’s α = 0.870 indicates a high degree of scale reliability [33].

### 2.3. Statistical Analysis

SPSS 25.0 and Excel software were used for data processing. The steps can be summarized as follows. (1) Preprocess the data and retest or reject the missing data. Then, re-score the reverse scoring items. (2) Descriptively analyze the data for MPA tendencies, physical exercise, and peer support of the university student respondents. Use the chi-square test to analyze the differences in the physical exercise levels of students of different genders and grades (effect size is based on Cramer’s Coefficients). Use η^2^ coefficient for the effect amount and perform a one-way ANOVA to analyze the differences in MPA and peer support between genders, students’ grades, and peer support. (3) Conduct a Kendall correlation analysis to test the correlation of MPA tendency (including withdrawal symptoms, highlighted behavior, social comfort, and mood changes) with physical activity level and peer support (including welcome, autism, and rejection) in university students. (4) Perform a linear hierarchical regression analysis to verify the predictive effect of MPA on physical exercise in university students and conduct a moderating effect test of peer relationships. Standardize the variables (Z score) before calculating the moderating effect.

## 3. Results

### 3.1. General Trends

As shown in Table 2, the rate of “low exercise” among males (54.4%) was prominently lower than that among females (85.5%); in other words, significantly more females than males engaged in low exercise. Physical exercise between males and females also differed significantly (*p* < 0.001, Cramer’s V = 0.359). In terms of the grade indicators, physical exercise varied among students across different grades (*p* < 0.001, Cramer’s V = 0.078).

Table 3 shows significant differences between males and females in terms of MPA (F = 2.748, *p* < 0.001, η^2^ = 0.031) and peer relationships (F = 7.454, *p* = 0.022, η^2^ = 0.234). The differences were also significant from the perspective of grade, except for exclusiveness (F = 0.329, *p* = 0.263).

### 3.2. Correlation Analysis

Table 4 shows the correlation between physical activity level and MPA, with dimensions between −0.262 and −0.556, presenting statistically significant differences (*p* < 0.001). The correlation between physical exercise level and peer relationships, with dimensions between 0.208 and 0.275, was also statistically significant (*p* < 0.001). The correlations between the total score of the peer relationships of students and each dimension were between 0.758 and 0.911, and the differences were statistically significant (*p* < 0.001). The correlations between peer relationships and MPA, with dimensions between −0.291 and −0.382, also showed significant differences (*p* < 0.001).

### 3.3. Regression Analysis

Table 5 presents the results of the moderating effect analysis of peer relationships. In model 1, the explanatory rate of the independent MPA and regulatory peer relationship variables was 12% for the dependent variable of physical exercise. Comparatively, the predictive power in model 2 increased by 3%. In other words, under the condition that the independent variable of MPA and the regulatory variable of peer relationships remain unchanged, the 4% increase can explain the predictive ability of the interaction term for the dependent variable of physical exercise, i.e., the contribution rate of the regulatory effect. Furthermore, the change in the significance of F was less than 0.001 in model 1 and 0.044 (i.e., <0.05) in model 2. These results indicate that the independent variable of MPA and the regulatory variable of peer relationships have a significant effect on the prediction of the dependent variable of physical exercise. Furthermore, the *p*-value of the ANOVA analysis was less than 0.001, which indicates that the regulatory effect of the regulatory variable of peer relationship on the independent variable of MPA was significant, and a regulatory effect existed. Therefore, peer relationships were a moderator of MPA and physical exercise.

## 4. Discussion

In the era of the mobile Internet, mobile phones occupy an increasingly important position in the study and life of university students. This study investigated and analyzed the physical exercise, MPA, and peer relationships of students in two universities in China, and it explored the intrinsic relationships of physical exercise, MPA, and peer relationships among university students.

The physical exercise level of university students was low, mainly manifesting as a low intensity of physical exercise, a small amount of exercise, a short exercise time, and a low frequency of exercise. The results are consistent with findings among U.S. university students by Wilson [34]. Studies have also shown that university students tend to choose indoor learning and entertainment and leisure rather than staying outdoors during leisure time to perform physical exercise activities viewed as beneficial to their physical and mental health [35]. This situation implies large amounts of static time in front of the screen (i.e., the time spent using mobile phones and other devices) and sedentary time, hence the increasingly low consumption of exercise and other energy activities, which is dramatically reducing the total amount of physical activity [36].

The exercise intensity, duration, and frequency of males were higher than those of females, which is consistent with the findings of previous studies [37]. The emergence of this phenomenon may be related to males having a higher interest in and awareness of sports than females. Sports often symbolize hard work and victory, which seem to be more in line with society’s expectations for male personality characteristics; this situation can explain why males appear to be more active than females. Meanwhile, quiet and docile features are the labeled roles that stereotypes give to females. Furthermore, females generally choose low-intensity physical activities (housework, walking, and jogging) or nonconfrontational ones (yoga and gymnastics) in their daily lives as the main approaches to spending their leisure time. By contrast, males prefer to engage in competitive sports (basketball, football, and tennis). Overall, when performing physical exercise, males are more active than females, and their activity level is higher.

The results of the correlational analyses indicate highly significant negative correlations between the physical exercise level of university students and the four dimensions of MPA (withdrawal symptoms, prominent behavior, social comfort, and mood changes). In other words, the greater the exercise intensity of university students, the longer their exercise time, the higher their frequency of participating in physical exercise, and the lower their likelihood of MPA. The physical exercise level of university students may also negatively affect their MPA; that is, the higher their physical activity level, the lower their MPA. Existing studies have shown that people with high usage of mobile phones tend to reduce their opportunities to engage in physical activity [38,39], and their sedentary behavior increases and calorie consumption decreases [1,40]. Therefore, colleges and universities should seriously consider implementing relevant measures to help students reduce mobile phone use and, if necessary, carry out psychological interventions, especially among university students, to reduce the probability of MPA.

Students’ peer relationships are positively correlated with physical activity level and negatively correlated with MPA. Peer relationships provide important context in the socialization process of university students. Healthy peer relationships can help avoid risks while providing a good social support environment for the physical and mental development of individuals [41]. Furthermore, with healthy relationships, peers can participate in physical exercise and develop social cognition and social skills through mutual encouragement and learning, among others. Individuals tend to participate in physical activity when accompanied by a companion or a close friend [23]. Peer support is positively correlated with an individual’s participation in physical activity, and support from friends can encourage physical activity. Moreover, individuals with better peer relationships likely have a stronger sense of self-identity, fulfillment, and motivation for voluntary participation [42] and, to a certain extent, provide a healthy alternative to continuous sitting/lying and low activity levels, which jointly reduce the likelihood of MPA. However, interestingly, negative peer relationships may reduce the enjoyment of individual participation in physical activity, eventually reducing the degree of physical participation and increasing the likelihood of MPA.

MPA significantly and negatively affects the physical activity of university students, which is consistent with previous views [20,38]. Mobile phone operations, such as online social networking, app shopping, and games, are recognized as static or sedentary behaviors in front of the screen [43]. Under normal circumstances, university students with serious addictions habitually ignore the surrounding environment, ignore real interpersonal communication, and easily indulge in the virtual network world mediated by mobile phones, which increases their static screen behavior, further affecting the execution and experience of daily physical activities. Such individuals are emotionally fickle, less stable, and suffer from anxiety, especially when experiencing withdrawal from mobile phone use [44]; they are also accustomed to focusing their lives on mobile phone network information acquisition or mobile phone technology operation, which increases sedentary behavior and interferes with normal physical activities. MPA is a type of inactive negative behavior that triggers a range of similarly negative psychological reactions (e.g., social exclusivity, procrastination in learning, and susceptibility to loneliness), increases the incidence of obesity, and reduces daily physical activity, and it can help to explain the low levels of energy expenditure in some university students [45,46].

Among university students whose physical exercise behavior is affected by MPA, the moderating effect of peer relationships is significant. According to social learning theory, the external environment has radiating and debugging effects on the psychology and behavior of a person [21]. In real life, when university students feel extremely distressed and troubled due to a lack of interpersonal security or discord in interpersonal relationships, they often expect to face or suffer from low social evaluation [47,48]. Such perceptions result in a series of negative psychological reactions (social exclusivity and self-isolation) and negative coping styles, eventually restricting the specific practice of physical exercise. The abovementioned phenomena may be attributed to a period of social adaptation and development among university students. In particular, the more negative life experiences triggered by external situations, the easier it is for university students to induce the self-preferential “empathy” of mobile online interaction to offset seemingly unobtainable self-esteem maintenance, psychological comfort, care, and emotional support in real-life situations. Therefore, poor peer relationships not only exacerbate the frequency of out-of-control dependence on mobile phone use but also likely produce serious MPA. Social support theory holds that emotions and care conveyed by the social environment stimulate a sense of social identity and self-identity among university students, leading to more dynamic and enthusiastic social behavior [21]. Therefore, harmonious peer relationships are an important factor in improving the sports exercise of university students.

Peer relationships are an important issue that every university student needs to face in their life on campus. On the basis of the analyzed data in this study, during interpersonal interaction, if university students perceive more serious trouble, then they are more likely to engage in self-isolation, social withdrawal, negative self-presentation, low satisfaction, and other psychological drawbacks, hence their higher inclination toward the negative impacts of stressors (competition, contradiction, and conflict). Simultaneously, university students tend to resist or reject positive social behavior and manifest physical activity states, such as low activity, slacking or laziness, procrastination, and avoidance [49]. Furthermore, self-determination theory holds that the failure to meet basic psychological needs, such as relationship needs, can lead to motivational externalization or even the absence of motivation, which inhibits the execution and maintenance of one’s activities [50]. Regarded as an important factor influencing the development of social adaptability and social skills among university students, peer relationships can change the thinking schema of how they understand the world, further affecting their choices, expressions, and experiences of social practices, either restricting or motivating their perceptions about physical exercise.

The limitations of our study should be acknowledged when interpreting the results. On the basis of relevant theories and the reviewed literature, this study adopted a cross-sectional research design to investigate the intrinsic relationship among the aforementioned variables. However, given the social and psychological factors of MPA and the complex interfering factors that likely manifested during the testing operation, the influence on each variable still needs to be demonstrated and tested via longitudinal research and quasi-experimental research, which is the direction of future work. Second, the selection of research objects and sampling methods needs to be further improved by expanding the survey scope and accurately selecting representative cities or university students as survey objects. Future work should not only increase the sample size but also consider the representativeness of the sample to verify the universality and rationality of the research conclusions. Finally, since MPATS has not yet completed the construction of a normal distribution of the university student population, only the raw scores of the scale were analyzed in this study, which may affect some of the interpretations of the results of this study.

## 5. Conclusions

The university students surveyed in this research engaged in medium- or low-level physical exercise. Furthermore, the more seriously the university students suffered from MPA, the lower their amount of physical exercise. The level of physical activity of males was higher than that of females. MPA and peer relationships were limiting factors in the physical exercise behaviors of university students. Under the lower effect of peer relationship regulation, MPA has a greater negative impact on physical exercise behavior. The data from this research can provide theoretical support to improve the participation of university students in physical activities. Good peer relationships and low cell phone use may be major factors in increasing physical activity in university students. In future studies, the dose–response relationship between variables can be verified through experimental design.

## Figures and Tables

**Figure 1 ijerph-20-02685-f001:**
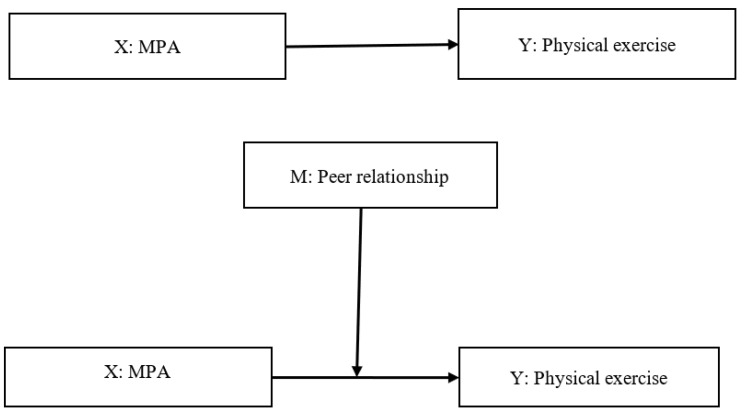
Conceptual model of this study.

**Table 1 ijerph-20-02685-t001:** Demographic characteristics of the study participants (n = 4959).

		n	%
Total			
		4959	100.0
Grade			
	Freshman	1708	34.4
	Sophomore	1752	35.3
	Junior	840	16.9
	Senior	659	13.3
Sex			
	Male	1878	37.9
	Female	3081	62.1

**Table 2 ijerph-20-02685-t002:** Status of physical exercise among university students.

			Descriptive		Statistical Values		
			n	%	χ^2^	*p*	Cramer’s V
Total							
		Low	3666	73.9			
		Middle	690	13.9			
		High	603	12.2			
Sex							
	Male (n = 1878)	Low	1022	54.4	637.825	<0.001	0.359
		Middle	401	21.4			
		High	455	24.2			
	Female (n = 3081)	Low	2644	85.8			
		Middle	289	9.4			
		High	148	4.8			
Grade							
	Freshman (n = 1708)	Low	1197	70.1	29.792	<0.001	0.078
		Middle	274	16.0			
		High	237	13.9			
	Sophomore (n = 1752)	Low	1349	77.0			
		Middle	228	13.0			
		High	175	10.0			
	Junior (n = 840)	Low	638	76.0			
		Middle	92	11.0			
		High	110	13.1			
	Senior (n = 659)	Low	482	73.1			
		Middle	96	14.6			
		High	81	12.3			

**Table 3 ijerph-20-02685-t003:** Current status of MPA and peer relationships among university students.

	Total		Male		Female		F	*p*	η^2^	Freshman		Sophomore		Junior		Senior		F	*p*	η^2^
	M	SD	M	SD	M	SD				M	SD	M	SD	M	SD	M	SD			
Withdrawal symptoms	15.962	5.891	15.999	6.142	15.939	5.734	2.037	0.004	0.013	15.490	5.321	16.296	5.812	16.394	6.564	15.745	6.488	7.359	<0.001	0.087
Highlight behavior	8.725	4.096	9.022	4.371	8.543	3.909	4.785	<0.001	0.107	7.812	3.442	9.192	4.111	9.419	4.632	8.962	4.466	45.846	<0.001	0.062
Social comfort	7.712	3.343	7.665	3.428	7.741	3.291	1.934	0.007	0.012	7.466	3.212	7.871	3.228	7.930	3.603	7.651	3.591	5.679	0.001	0.044
Mood alteration	6.924	3.172	6.953	3.312	6.906	3.084	2.281	<0.001	0.013	6.452	2.816	7.164	3.150	7.350	3.535	6.964	3.456	21.303	<0.001	0.172
MPATS	39.322	15.139	39.638	16.047	39.130	14.557	2.748	<0.001	0.031	37.220	12.988	40.523	15.075	41.093	17.259	39.322	16.896	18.677	<0.001	0.145
Welcoming	11.953	2.987	12.104	3.199	11.861	2.846	1.316	0.156	0.026	11.734	2.862	12.076	2.975	12.206	3.070	11.868	3.185	6.260	<0.001	0.040
Autism	17.743	3.819	17.922	4.025	17.634	3.684	2.047	0.004	0.035	17.575	3.664	17.913	3.782	17.874	3.920	17.560	4.146	3.093	0.026	0.026
Exclusiveness	14.326	2.111	14.444	2.215	14.255	2.041	1.452	0.087	0.016	14.320	2.121	14.366	2.082	14.368	2.116	14.184	2.151	0.329	0.263	0.001
PRS	44.022	7.735	44.470	8.158	43.749	7.454	1.733	0.022	0.234	43.629	7.714	44.356	7.660	44.448	7.572	43.612	8.131	4.027	0.007	0.154

**Table 4 ijerph-20-02685-t004:** Results of the correlation analysis.

	Withdrawal Symptoms	Highlight Behavior	Social Comfort	Mood Alteration	MPATS	Welcoming	Autism	Exclusiveness	PRS
Withdrawal symptoms									
Highlight behavior	0.804 **								
Social comfort	0.742 **	0.697 **							
Mood alteration	0.795 **	0.805 **	0.699 **						
MPATS	0.944 **	0.907 **	0.852 **	0.895 **					
Welcoming	−0.285 **	−0.329 **	−0.376 **	−0.310 **	−0.348 **				
Autism	−0.236 **	−0.331 **	−0.294 **	−0.303 **	−0.317 **	0.506 **			
Exclusiveness	−0.235 **	−0.329 **	−0.297 **	−0.298 **	−0.312 **	0.519 **	0.794 **		
PRS	−0.291 **	−0.382 **	−0.373 **	−0.353 **	−0.377 **	0.758 **	0.911 **	0.870 **	
PARS-3	−0.262 **	−0.256 **	−0.311 **	−0.269 **	−0.279 **	0.275 **	0.209 **	0.208 **	0.209 **

** stands for *p* < 0.001.

**Table 5 ijerph-20-02685-t005:** Summary of the results of the moderating effect.

Model	Model Summary					ANOVA	
	R^2^	ΔR^2^	df 1	df 2	Sig. F Change	F	*p*
1	0.112	0.120	2	3130	<0.001	36.610	<0.001
2	0.112	0.030	1	3129	0.044	31.071	<0.001

## Data Availability

The raw data supporting the conclusions of this article can be made available by the authors, without undue reservation.

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
