# Peer review of "Effect of Mobile Phone Addiction on Physical Exercise in University Students: Moderating Effect of Peer Relationships"

_ijerph, 2023, doi:10.3390/ijerph20032685_

Round 1

Reviewer 1 Report

The manuscript investigated the influence of mobile phone addiction (MPA) on physical exercise in college students, and peer relationships were introduced as a moderating variable. For better understanding, some modifications are suggested.

1) First, some references are written in Chinese and I recommend to translate those to English format for international readers.

2) Particularly, the references of MPA tendancy scale (MPATS) and Physical Activity Rating Scale (PARS-3) are in Chinese and I could not find the scales in English form. So, I suggest to include the scale questions in Appendix for clear understanding.

3) p.4 line 166-167 : “Complete conformity” to “complete nonconformity” are recorded as L as four points. : I can't understand the L. please explain more detail.

4) p.6 Table 4 : Table 4 includes all r and p values that make the table too complex. I suggest to exclude p values and indicate the statistically significance by using * (p<0.05) and **(p<0.01) in upper subscript form.

5) p.7 line 260: By contrast, males prefer to engage in competitive and competitive sports (basketball, football, and tennis). : => duplicate of ‘competitive’ ? or missing other words ?

Author Response

Point-by-point Responses to Reviewer 1

Dear reviewer,

Thank you for the time and effort that you have dedicated to providing your insightful and valuable comments on our manuscript. Although I do not know the situation around you, please stay healthy and keep safe.

Sincerely,

Comment 1:

First, some references are written in Chinese and I recommend to translate those to English format for international readers.

Response 1:

Thank you for your comment. We have conducted a format check for papers in accordance with MDPI's submission requirements.

Comment 2:

Particularly, the references of MPA tendancy scale (MPATS) and Physical Activity Rating Scale (PARS-3) are in Chinese and I could not find the scales in English form. So, I suggest to include the scale questions in Appendix for clear understanding.

Response 2:

Thank you for your comment. We have added the scale to the attachment. It should be noted that the original of MPATS is Chinese, and the original of PARS-3 is in Japanese.

Comment 3:

p.4 line 166-167 : “Complete conformity” to “complete nonconformity” are recorded as L as four points. : I can't understand the L. please explain more detail.

Response 3:

Thank you for your comment. There is an error in the formulation here, which we have revised in the text. The options on the scale range from " Complete conformity " to "completely nonconformant", assigning values of 1 to 4 points.

Comment 4:

p.6 Table 4 : Table 4 includes all r and p values that make the table too complex. I suggest to exclude p values and indicate the statistically significance by using * (p<0.05) and **(p<0.01) in upper subscript form.

Response 4:

Thank you for your comment. This is a very good suggestion. We have modified it as you suggested.

Comment 5:

p.7 line 260: By contrast, males prefer to engage in competitive and competitive sports (basketball, football, and tennis). : => duplicate of ‘competitive’ ? or missing other words ?.

Response 5:

Thank you for your comment. The‘competitive’ here is a duplicate. We have made modifications.

Thank you so much for letting me learn a lot of new knowledge!

Reviewer 2 Report

see file attached

Author Response

Point-by-point Responses to Reviewer 2

Dear reviewer,

Thank you for the time and effort that you have dedicated to providing your insightful and valuable comments on our manuscript. Although I do not know the situation around you, please stay healthy and keep safe.

Sincerely,

Comment 1:

In the measurements section, you describe the MPATS. Please add a closer description of the interpretation of the attainable scores. 80 points is the highest value, which stands for a high MPA. But what are the thresholds for low, moderate, or high addiction? This is an important information for understanding the values of your sample (see comment number 8).

Response 1:

Thank you for your comment. This is indeed a very important question. But unfortunately, MPATS does not currently have a measurement norm for college students, which is the threshold you mentioned. Therefore, in the introduction of the article, we only analyzed the measured raw scores of MPATS. And we explain it in limitations. Thanks again for your comment.

Comment 2:

In l156 you talk about calculating about the PA-score through “public announcement”. What is this, I don’t understand the term, I suppose. If you mean the formula for calculating the value in l163, please find another expression and reformat the formula.

Response 2:

Thank you for your comment. Thanks for pointing out. There is a mistake in the expression here, it should be "equation".

Comment 3:

table 3 is very complex. Is it really necessary to specify the data for each subgroup? You do not take this into account later (especially the grades). The same is true for Table 2, where the values of the different grades are very similar, so it is not really necessary to present them in a table. One sentence in the descriptive part (ll 194ff) would suffice.

Response 3:

Thank you for your comment. Tables 3 and 4 exist to show whether there are differences in measurements in terms of gender and grade. For the format, we use the specific requirements of MDPI. Thanks again for your comment.

Comment 4:

able 4 should be shortened. The entire lower half could be omitted, since the information is all contained in the upper part.

Response 4:

Thank you for your comment. We very much agree with your suggestion. And we have already revised it in the paper.

Comment 5:

please add the information about the variables included in your regression analysis under table 5 (and in the following text you should describe more concretely which variables influence which variables. Please do not just talk in general terms about dependent and independent variables, but name the concrete contents (MPA, PA, peerrelationships). This makes it easier to understand and is important because you want to prove causality with this calculation).

Response 5:

Thank you for your comment. This is a very good comment. We have modified it as you prompted. Thanks again for your comment.

Comment 6:

You argue a lot with the significant correlations or differences between groups and infer causality based on regression analysis. This is not wrong in principle, but do the data really show this? Significant differences are not really meaningful with this sample size. Please calculate and report the effect size.

Response 6:

Thank you for your comment. We have modified it as you suggested. Thanks again for your comment.

Comment 7:

Why did you not use a path analysis or a SEM? This could provide much more robust support for the assumed causality than correlation analyses and regression.

Response 7:

Thank you for your comment. A very good suggestion. We use this approach for several reasons. First, this study adopts a cross-sectional study design, and the discussion of causality may not be discussed in this study. Statistically speaking, this study discusses the influencing factors of PA and tends to predict. Second, such methods are used only to discuss whether peer relationships are moderators. The intensity and method of moderating the effect are beyond the scope of this study. Finally, there is a Chinese proverb " A black cat is a white cat, He is a good cat that catches mice”. We believe that the richness of the methodology is reflected in the richness of the methodology by obtaining the final result through different methods.

Comment 8:

To return to the first comment. The MPATS score is 40 for all subgroups. Does that even represent the addiction you are reporting? 40 out of a maximum of 80 points does not seem exceptionally high to me. Could you please interpret these values qualitatively more clearly and explain what they mean? It almost seems a little overinterpreted to me. The addiction may not be as high as you claim.

Response 8:

Thank you for your comment. Indeed, as you said, it is difficult to define behavior without norms. This is indeed the shortcoming of this study. Therefore, in this study, only the raw scores of MPATS were analyzed, and the degree of addictive behavior was not discussed.

Comment 9:

Finally, the conclusion section is not really about conclusions. It’s more a summary of the whole study. Please present in a comprehensible way what conclusions should be drawn from the results of the study. What should change? What adjustments or changes should be addressed in the future, by whom and how, in order to achieve what?

Response 9:

Thank you for your comment. We have modified it as you suggested. Thanks again for your comment.

Thank you so much for letting me learn a lot of new knowledge!

Reviewer 3 Report

The review results for this manuscript are as follows.

1.      For university students, the rationale for the hypothesis that the PA has a positive effect on the MPA should be described. To this, the basis for applying peer relationships as an mediating variable should be added.

2.      Describe the design of this study.

3.      Describe the criteria for selection and exclusion of study subjects.

4.      Describe the basis for calculating the number of samples through G*Power.

5.      Specific descriptions of the operational definition of physical activity to be measured in this study, that is, the type, period, environment, guider, and measurer of body displacement, are needed.

6.      Add the operational definition of peer relationships, the mediating variable in this study.

7.      The bias effects of concern in this study should be discussed.

Author Response

Point-by-point Responses to Reviewer 3

Dear reviewer,

Thank you for the time and effort that you have dedicated to providing your insightful and valuable comments on our manuscript. Although I do not know the situation around you, please stay healthy and keep safe.

Sincerely,

Comment 1:

For university students, the rationale for the hypothesis that the PA has a positive effect on the MPA should be described. To this, the basis for applying peer relationships as an mediating variable should be added.

Response 1:

Thank you for your comment. A very good suggestion, thanks again for your comment. We try to reply, hoping to get your approval. First of all, this study explores the impact of MPA on PA, and its significance lies in exploring some influencing factors that promote PA, so the analysis of the positive effects of PA on MPA is not too much elaborated in the original article. Second, this study focuses on the moderating effect of peer relationships, answering the question of "when" MPA affects PA, and "how" of this effect. It is not a question of the MPA "why" and "how" affects the PA. This may be distinguished from the mediation effect test. Therefore, the questions you asked may not be too much in the study. I think this may not be addressed by the study design of this study, so we describe the limitations of this study in Limitations. We hope that our response will be recognized by you.

Comment 2:

Describe the design of this study.

Response 2:

Thank you for your comment. We have added the study design of this study to the methodological part of the paper(L127-L130). Please refer to the original text.

Comment 3:

Describe the criteria for selection and exclusion of study subjects.

Response 3:

Thank you for your comment. We have added the criteria of this study to the methodological part of the paper(L147-L155). Please refer to the original text.

Comment 4:

Describe the basis for calculating the number of samples through G*Power..

Response 4:

Thank you for your comment. We have added the number of samples of this study to the methodological part of the paper(L156-L162). Please refer to the original text. By the way, our sample size is much larger than the minimum sample size.

Comment 5:

Specific descriptions of the operational definition of physical activity to be measured in this study, that is, the type, period, environment, guider, and measurer of body displacement, are needed.

Response 5:

Thank you for your comment. A very good suggestion. In order to show the data collection of physical exercise in this study more clearly. We present the questionnaire as an Appendix. This makes it clearer and easier for the reader to read.

Comment 6:

Add the operational definition of peer relationships, the mediating variable in this study.

Response 6:

Thank you for your comment. We have added relevant argument(L90-L102).

Comment 7:

The bias effects of concern in this study should be discussed.

Response 7:

Thank you for your comment. For bias effects, we have to admit that it does exist, so we focused on the limitations.

Thank you so much for letting me learn a lot of new knowledge!

Reviewer 4 Report

The authors submitted an article on the effect of mobile phone addiction on physical exercise in college students: moderating effect of peer relationships. The topic is very interesting and a current issue in our globalized society.

The article is well written and has an adequate structure, but some parts need amendments to make the article stronger:

1.     First of all, I would change “college students” for “university students”.

2.     Abstract:

L18-19: I would put this sentence at the end of the abstract and add why the data from your research can provide support to improve students’ participation in PA.

3.     Keywords: It is strongly recommended not to repeat the keywords and the words of the title of your article.

4.     Introduction:

L51-52: Why is it so important to investigate the effect of MPA on PA among those students with behavioral self-management ability problems? Could you strengthen this statement, please?

5.     Materials and Methods:

L115: I suggest modifying table 1 (improve design) and insert it at L128.

L151: I’m missing a short explanation of this revision process of PARS-3 in Chinese. What about the cultural issue translating and adapting a Japanese Scale into Chinese? Different cultures? Validation of PARS-3? How many items? Where they all retained from original Japanese version?

L163: align the line with text

L171: alpha Cronbach of factors are not very high. Why?

6.     Results:

Leave an introductory line before starting with the tables or…

L193: I suggest moving table 2 after the text (now lines 194-199)

Modify table, leaving a line between each block of “Low, medium, high”

L200: Please modify table table3 or maybe consider splitting it into two different tables to make it clearer. Move table 3 after the text (now lines 201 – 204). Idem table 4 and 5.

7.     L231 and Discussion:

What about the MPA in the peers? If they are also addicted to their MP in China, what then? What is the global % of MPA among university students? in general population and children?

L255-256: In which culture? Not in all! Other countries?

8.     Discussion:

L357: please specify or introduce a limitation section

9.     Conclusions:

To my opinion, the conclusion section is too short. Please add on theoretical and practical implications.

10.  References: please follow MDPI Guidelines

Provide translation of Chinese or other non-English references

Author Response

Point-by-point Responses to Reviewer 4

Dear reviewer,

Thank you for the time and effort that you have dedicated to providing your insightful and valuable comments on our manuscript. Although I do not know the situation around you, please stay healthy and keep safe.

Sincerely,

Comment 1:

First of all, I would change “college students” for “university students”.

Response 1:

Thank you for your comment. Very good advice. Agree! We have already made the replacement.

Comment 2:

Abstract: L18-19: I would put this sentence at the end of the abstract and add why the data from your research can provide support to improve students’ participation in PA.

Response 2:

Thank you for your comment. We very much agree with your suggestion and have rewritten the abstract.

Comment 3:

Keywords: It is strongly recommended not to repeat the keywords and the words of the title of your article.

Response 3:

Thank you for your comment. Strongly agree. We have made modifications.

Comment 4:

Introduction: L51-52: Why is it so important to investigate the effect of MPA on PA among those students with behavioral self-management ability problems? Could you strengthen this statement, please?

Response 4:

Thank you for your comment. We've added a description based on your comments(L54-L59). We look forward to resolving your doubts.

Comment 5:

Materials and Methods: L115: I suggest modifying table 1 (improve design) and insert it at L128.

Response 5:

Thank you for your comment. Thank you very much for your advice. We tried our best to improve and made the move in the format as you suggested.

Comment 6:

L151: I’m missing a short explanation of this revision process of PARS-3 in Chinese. What about the cultural issue translating and adapting a Japanese Scale into Chinese? Different cultures? Validation of PARS-3? How many items? Where they all retained from original Japanese version?

Response 6:

Thank you for your comment. PARS-3 is widely used in the current research on physical exercise behavior in China. Possible reasons are that PARS-3 is short and easy to understand. The degree of Chineseization of PARS-3 has been relatively mature, and the relevant introduction work has been completed. In order to show the procedure of PARS-3 more clearly, we will place PARS-3 as an attachment in the paper. At the same time, reference No. 31 was completed by our team, and you are welcome to refer to it. Among them, the introduction process of PARS-3 is more detailed. Hope our response will solve your doubts.

Comment 7:

L163: align the line with text.

Response 7:

Thank you very much! Regarding format-related issues, we have sorted out the relevant requirements of MDPI.

Comment 8:

L171: alpha Cronbach of factors are not very high. Why?

Response 8:

Thank you for your comment. I think it shouldn't be too low, and this reliability is acceptable. We contacted Zhang, the person who revised the scale. She believes that the possible reason is that "lengthy questions will increase the irritability of the participant, thereby reducing the reliability of the scale." It is true that the questionnaire requires a large number of questions. This is probably one of the main reasons.

Comment 9:

Results: L193: I suggest moving table 2 after the text (now lines 194-199)

Response 9:

Thank you very much! Regarding format-related issues, we have sorted out the relevant requirements of MDPI.

Comment 10:

Modify table, leaving a line between each block of “Low, medium, high”

Response 10:

Thank you for your comment. We have modified it as you suggested.

Comment 11:

L200: Please modify table table3 or maybe consider splitting it into two different tables to make it clearer. Move table 3 after the text (now lines 201 – 204). Idem table 4 and 5.

Response 11:

Thank you for your comment. For Table 3, we recommend placing it horizontally and we will communicate with the editor. Regarding format-related issues, we have sorted out the relevant requirements of MDPI.

Comment 12:

L231 and Discussion: What about the MPA in the peers? If they are also addicted to their MP in China, what then? What is the global % of MPA among university students? in general population and children?

Response 12:

Thank you for your comment. Unfortunately, the current study may not be able to answer your question for the time being. First, the scale A used in this study does not currently carry out norm construction, so it is not possible to analyze the degree of addiction formation. Secondly, in view of the current diversity of MPA measurements, there are no relevant reports to comprehensively analyze the world situation of MPA, which is also a problem that future scholars need to explore. Unfortunately, although we have read a lot of literature, the relevant research is indeed relatively lacking.

Comment 13:

Discussion: L357: please specify or introduce a limitation section

Response 13:

Thank you for your comment. We've made additions.

Comment 15:

Conclusions: To my opinion, the conclusion section is too short. Please add on theoretical and practical implications.

Response 15:

Thank you for your comment. We've made additions.

Comment 16:

References: please follow MDPI Guidelines. Provide translation of Chinese or other non-English references.

Response 16:

Thank you for your comment. Regarding format-related issues, we have sorted out the relevant requirements of MDPI.

Thank you so much for letting me learn a lot of new knowledge!

Round 2

Reviewer 2 Report

Thank you for answering and reacting to my queries. I still think, the interpretation of the data is a little bit too speculative, but over all, the study shows interesting insights in an actual topic

Reviewer 3 Report

This manuscript has been properly reviseded according to the reviewer's comments. It is hoped that the results of this study will be used for college students who are addicted to smartphones. Thank you for your efforts.

Reviewer 4 Report

Dear authors: thank you for answering all my questions and the amendments.

I would like to suggest to publish your article in its presetn form .

Thank you for your interesting contribution.